# Interaction of the pioneer transcription factor GATA3 with nucleosomes

Hiroki Tanaka[1,2,6], Yoshimasa Takizawa[1,6], Motoki Takaku [ID] [3,4,6], Daiki Kato[2,5], Yusuke Kumagawa[1], Sara A. Grimm[3], Paul A. Wade [ID] [3✉] & Hitoshi Kurumizaka [ID] [1,2✉]

During cellular reprogramming, the pioneer transcription factor GATA3 binds chromatin, and in a context-dependent manner directs local chromatin remodeling and enhancer formation. Here, we use high-resolution nucleosome mapping in human cells to explore the impact of the position of GATA motifs on the surface of nucleosomes on productive enhancer formation, finding productivity correlates with binding sites located near the nucleosomal dyad axis. Biochemical experiments with model nucleosomes demonstrate sufficiently stable transcription factor-nucleosome interaction to empower cryo-electron microscopy structure determination of the complex at 3.15 Å resolution. The GATA3 zinc fingers efficiently bind their target 5′-GAT-3′ sequences in the nucleosome when they are located in solvent accessible, consecutive major grooves without significant changes in nucleosome structure. Analysis of genomic loci bound by GATA3 during reprogramming suggests a correlation of recognition motif sequence and spacing that may distinguish productivity of new enhancer formation.

[1] Laboratory of Chromatin Structure and Function, Institute for Quantitative Biosciences, The University of Tokyo, 1-1-1 Yayoi, Bunkyo-ku, Tokyo 113-0032, Japan. [2] Graduate School of Advanced Science and Engineering, Waseda University, 2-2 Wakamatsu-cho, Shinjuku-ku, Tokyo 162-8480, Japan. [3] Epigenetics and Stem Cell Biology Laboratory, National Institute of Environmental Health Sciences, Research Triangle Park, Durham, NC 27709, USA. [4] Department of Biomedical Sciences, University of North Dakota School of Medicine and Health Sciences, Grand Forks, ND 58202, USA. [5] Present address: Laboratory for Drug Discovery, Pharmaceuticals Research Center, Asahi Kasei Pharma Corporation, 632-1 Mifukulzunokuni-shi, Shizuoka 410-2321, Japan. [6] These authors contributed equally: Hiroki Tanaka, Yoshimasa Takizawa, Motoki Takaku. ✉email: wadep2@niehs.nih.gov; kurumizaka@iqb.u-tokyo.ac.jp

Formation of new enhancers to activate transcriptional programs downstream of the signaling events of development, cellular reprogramming, and response to the environment require penetration of the chromatin barrier. A special class of transcription factors, termed pioneer transcription factors, has the capacity to bind nucleosomal DNA and nucleate the formation of a new enhancer[1,2]. The GATA family of proteins are zinc finger transcription factors that have established pioneering functions in multiple developmental pathways, including mammary epithelial cell differentiation[3–5]. GATA3 functions to establish new cell fates in cellular reprogramming systems[6–8] including driving of a mesenchymal to epithelial transition in breast cancer cell lines[9,10]. In these processes, GATA3 is proposed to bind target DNA sequence wrapped tightly within a positioned nucleosome and to induce chromatin remodeling, probably by recruiting a nucleosome remodeler[11].

GATA proteins use zinc finger motifs as DNA-binding domains[12,13]. The consensus recognition motif derived for GATA3, WGATAR, is believed to be engaged by zinc finger 2[14]. However, elegant structural studies of the GATA3 zinc fingers complexed with its target DNA revealed that the two zinc fingers bind to two neighboring 5′-GAT-3′ sequences located within the same major groove of the DNA[15]. Should this DNA-binding specificity be encountered on a nucleosome surface, one of the two zinc fingers would sterically clash with the histone core in the nucleosome. Therefore, the details by which the GATA3 zinc fingers bind their cognate recognition motif within a nucleosomal context remains poorly understood.

In our previous work[11], we expressed GATA3 at physiologic levels in a GATA3-negative breast cancer cell line, tracking GATA3 binding across the genome by chromatin immunoprecipitation-sequencing (ChIP-seq) and chromatin accessibility using the assay for transposase-accessible chromatin using sequencing (ATAC-seq). We identified three scenarios for GATA3 interaction with the chromatin fiber (Fig. 1a). In the most abundant category of loci identified (termed G2), GATA3 bound to transposase-accessible DNA, which remained accessible following GATA3 expression. At ~¼ of loci, GATA3 bound to inaccessible DNA, which became accessible following binding (termed G1 sites). Finally, at ~¼ of GATA3-bound loci, binding occurred at inaccessible sites that remained inaccessible following GATA3 expression (termed G3 sites).

Here we perform high-resolution nucleosome mapping in vivo in our cellular reprogramming system and assess the location of GATA3-binding sites relative to nucleosome positions in all three categories of GATA3 sites. We then use predictions from this analysis to design biochemical experiments to test the outcome of moving the GATA3-binding motif to different locations on the surface of a model nucleosome. We utilize one of the resulting GATA3–nucleosome complexes to determine the cryo-electron microscopy structure of the nucleosome complexed with the human GATA3 zinc fingers at 3.15 Å resolution. In the structure, the GATA3 zinc fingers bind their target DNA sequences in the nucleosome without significant deformation of the nucleosome structure. The zinc fingers of GATA3 bind two 5′-GAT-3′ sequences located in consecutive major grooves that are exposed to the solvent. Finally, we use the information derived from the structural analysis to reanalyze our genomic data, finding that the spacing of 5′-GAT-3′ motifs at GATA3 bound sites in vivo is non-random and that the strand orientation of these sequences is likewise non-random. Thus, the engagement of both GATA3 zinc fingers with DNA sequence in a specific spatial arrangement on the surface of nucleosomes may be key in the outcome of GATA3 binding to chromatin.

## Results

**GATA3 interaction with nucleosomes in vivo.** To assess local nucleosome architecture at GATA3-bound sites in our cellular reprogramming system[11], we employed micrococcal nuclease to map nucleosome boundaries. We selected 109 loci from G1 and 100 loci from G2 and G3, respectively, and used capture probes to 1500 bp regions centered on the GATA3 ChIP-seq peak to enrich from micrococcal nuclease-sequencing (MNase-seq) libraries (Fig. 1b). The resulting data provided substantial sequencing depth at the selected loci (Supplementary Fig. 1a, b), permitting the analysis of nucleosome positions before and after GATA3 expression at individual loci in the human genome. Examples of smoothed nucleosome positions[16] from each category are depicted along with GATA3 localization and ATAC accessibility (Fig. 1c–e). Some loci present a well-defined pattern of nucleosomes, whereas other loci have less defined nucleosome locations within the population sampled (Supplementary Fig. 1c).

We next defined the consensus GATA3 motif, WGATAR, closest to the highest GATA3 ChIP-seq signals within each peak (indicated as dashed squares in Fig. 1c–e) and mapped nucleosome dyad frequency across all captured loci within each class (see "Methods," Fig. 1f, g). As expected, at G2 loci nucleosome dyads were more frequently located away from the GATA3-binding motif, both before and after protein expression (Fig. 1f, g). This pattern is consistent with GATA3 binding to accessible DNA near a nucleosome, with subsequent widening of the nucleosome-depleted region (Fig. 1d). In contrast, at G1 loci we observed a striking enrichment of nucleosomal dyads at symmetrically located positions approximately two helical turns from the GATA3 motif prior to GATA3 expression, suggesting that the binding motif was preferentially located on the nucleosome surface at position superhelical location 2 (SHL2). After GATA3 expression, we observed nucleosomal dyads accumulated at distances in excess of 100 bp from the GATA motif (Fig. 1f, g), consistent with local chromatin remodeling resulting in movement of the nucleosome off the GATA motif (Fig. 1c). Finally, at G3 loci we observed very little position preference for the GATA motif prior to GATA3 expression (Fig. 1e and Supplementary Fig. 1c) with enrichment for nucleosomes positioning the binding site on the nucleosome surface at a position near the nucleosome edge (five helical turns from the dyad axis, position SHL5) after GATA3 expression (Fig. 1f, g). These results demonstrate that the context of GATA motifs in chromatin differs across the three classes of binding site, correlating with outcome.

**The zinc fingers of GATA3 bind nucleosomal DNA.** We turned to a biochemical system to further explore the interaction of GATA3 with nucleosomal DNA. GATA proteins recognize duplex DNA within the 5′-GAT-3′ sequences by their tandem zinc finger domains[15]. We first tested whether GATA3 binds the 5′-GAT-3′ sequences incorporated into nucleosomes at different translational and rotational positions using a 145 bp Widom 601 positioning sequence[17] as the template DNA. We chose this DNA sequence as an initial model to take advantage of the well-characterized positioning, thermodynamic stability and structural data available for this nucleosome[17–19]. Nucleosomes were then reconstituted with a GATA-binding motif, 5′-AGATANCATCT-3′ (5′-GAT-3′ sequences are underlined), in ten different positions around SHLs 2 and 5 (named SHL2a, SHL2b, SHL2c, SHL2d, SHL2e, SHL5a, SHL5b, SHL5c, SHL5d, and SHL5e; Fig. 2a red arrows and Supplementary Fig. 2a, b). In addition, the Widom 601 sequence contains three 5′-GAT-3′ sequences (Fig. 2a, blue arrows). The tandem zinc finger domain of human

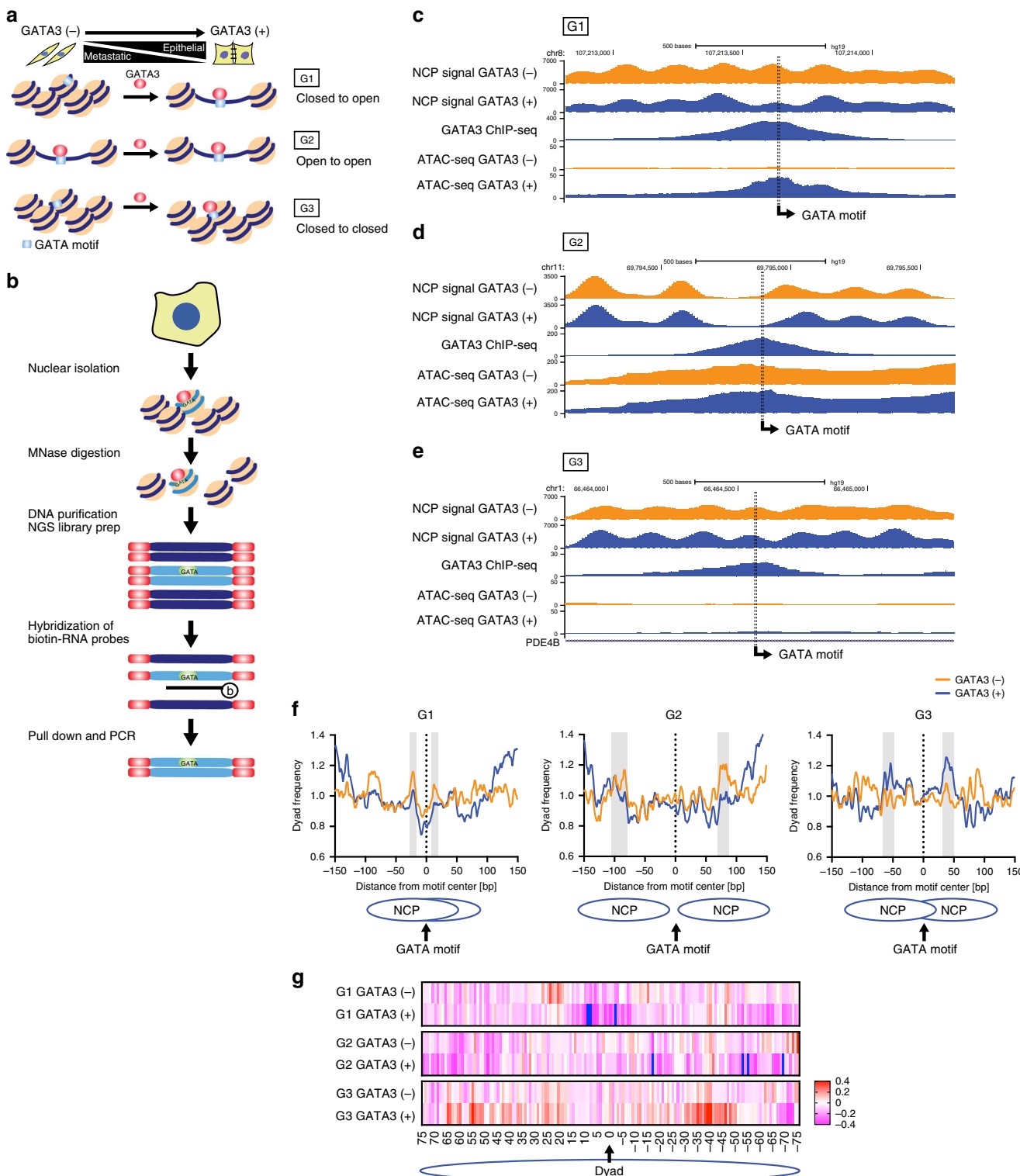

**Fig. 1 Nucleosome positioning at GATA3-binding peaks. a** GATA3 peak classification based on chromatin architecture. **b** Scheme of capture MNase-seq. Targeted GATA3 peaks were enriched with biotinylated RNA probes. Smoothed nucleosome signals (NCP signal) were calculated by iNPS[16]. **c-e** Genome browser view of capture MNase-seq data at G1 (**c**), G2 (**d**), and G3 (**e**). The dashed squares indicate the closest GATA motifs (from the highest ChIP-seq signals). **f** Metaplot showing averaged dyad (fragment center/midpoint) frequency in each GATA3 peak group obtained by capture MNase-seq. Dyad frequency was normalized by the capture MNase-seq data from sonicated DNA fragments. Ellipses indicate enriched symmetric positioning of nucleosomes in each peak class. **g** Heatmap showing enrichment of preferred GATA3 motif positioning on the nucleosome. Frequency of observed GATA3 motif positioning within the nucleosome was normalized by the capture MNase data from the sonicated DNA fragments and the calculated fold changes in each peak group and cell line are indicated as log2 values. Only one GATA3 motif that is closest from the highest GATA3 ChIP-seq signal intensity was chosen from each peak.

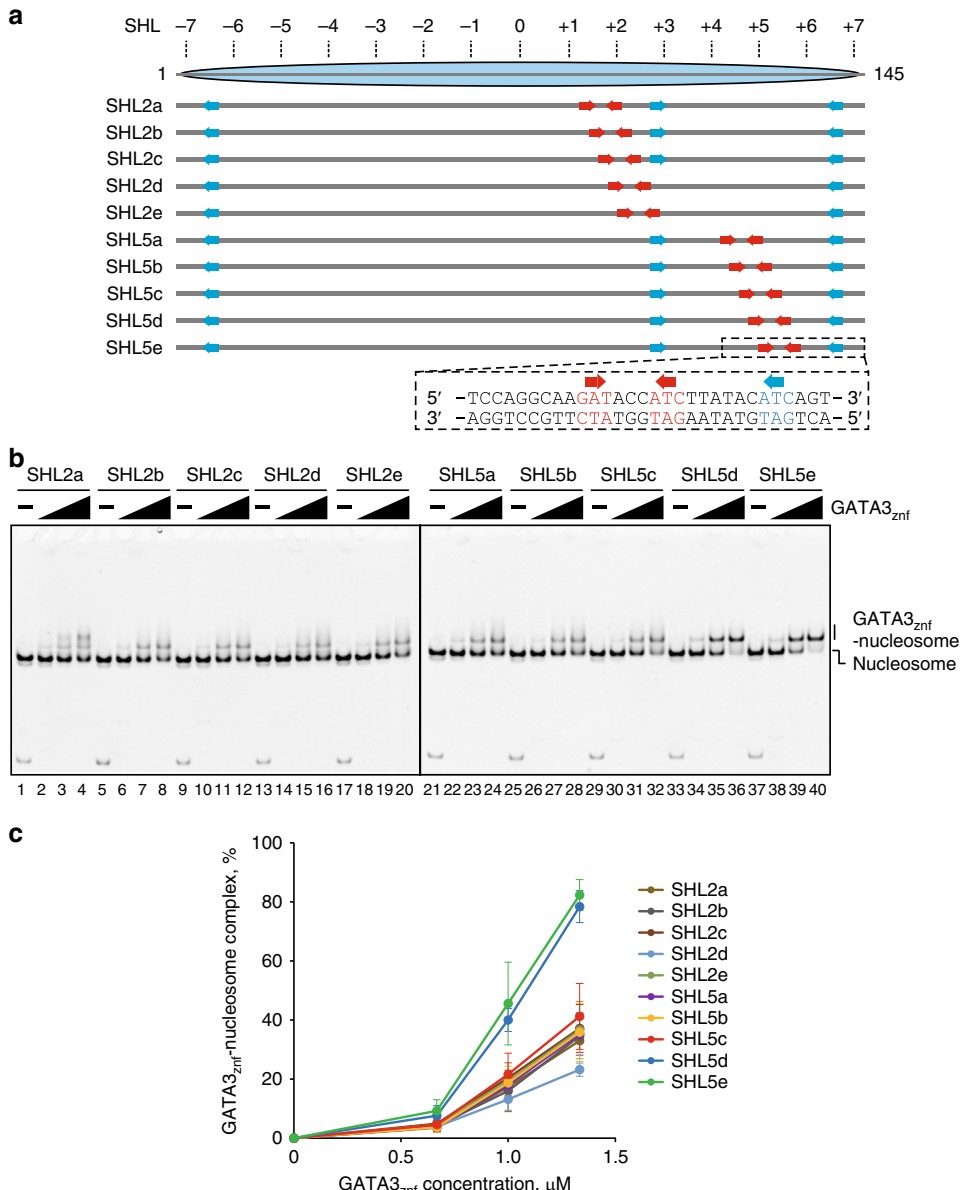

**Fig. 2 GATA3$_{znf}$ binds nucleosomal target DNA. a** Design of the GATA3 target nucleosomes. Red arrows indicate the introduced 5'-GAT-3' (or 5'-ATC-3') sequences. Blue arrows indicate 5'-GAT-3' (or 5'-ATC-3') sequences originally contained in the Widom 601 sequence. Right arrows indicate 5'-GAT-3' sequence. Left arrows indicate 5'-ATC-3' sequence. **b** Electrophoretic mobility shift assay (EMSA) of the GATA3 target nucleosomes with GATA3$_{znf}$. The nucleosome (0.33 μM) was mixed with GATA3$_{znf}$ (0, 0.67, 1.0, and 1.33 μM). The reaction mixtures were incubated at 25 °C for 30 min and were then analyzed by native-PAGE. The gel was stained with ethidium bromide. Lanes 1–4, 5–8, 9–12, 13–16, 17–20, 21–24, 25–28, 29–32, 33–36, and 37–40 indicate results for the nucleosomes containing SHL2a, SHL2b, SHL2c, SHL2d, SHL2e, SHL5a, SHL5b, SHL5c, SHL5d, and SHL5e, respectively. Three independent experiments were performed, and the reproducibility was confirmed (Supplementary Fig. 3). **c** Quantification of the results shown in **b**. The average % values of three independent experiments shown in **b** and Supplementary Fig. 3 were plotted against the GATA3$_{znf}$ concentration. Data are represented as mean values ± SD (*n* = 3 independent biological replicates). Source data are provided as a Source Data file.

GATA3 (GATA3$_{znf}$) was purified as a recombinant protein (Supplementary Fig. 2c).

We tested the GATA3$_{znf}$ binding to the nucleosome by an electrophoretic mobility shift assay (EMSA). Specific bands corresponding to the GATA3$_{znf}$–nucleosome complexes were observed with all nucleosomes containing the GATA consensus sequence in different nucleosomal locations (Fig. 2b and Supplementary Fig. 3a, b). Unexpectedly, GATA3$_{znf}$ also generated a specific band with a nucleosome with no added GATA consensus sequence (Fig. 3a lanes 1–4, Fig. 3b, and Supplementary Fig. 4a, b). However, this specific binding was no longer observed when all the intrinsic 5'-GAT-3' sequences of the

Widom 601 sequence were disrupted (Fig. 3a lanes 9–12, Fig. 3b, and Supplementary Fig. 4a, b). Binding of the zinc finger fragment to the nucleosome surface resulted in specific binding events that were substantially stronger near the nucleosome periphery than near the dyad axis (Figs. 2b, c and Supplementary Fig. 3a, b). The rotational setting of the GATA motif had little influence on binding affinity near the nucleosomal dyad axis around position SHL2 (Fig. 2c), while binding increased substantially as rotational setting changed near the edge of the nucleosome around position SHL5 (Fig. 2c). These results indicate that GATA3$_{znf}$ specifically binds the 5'-GAT-3' sequences within the model nucleosome and highlights inherent

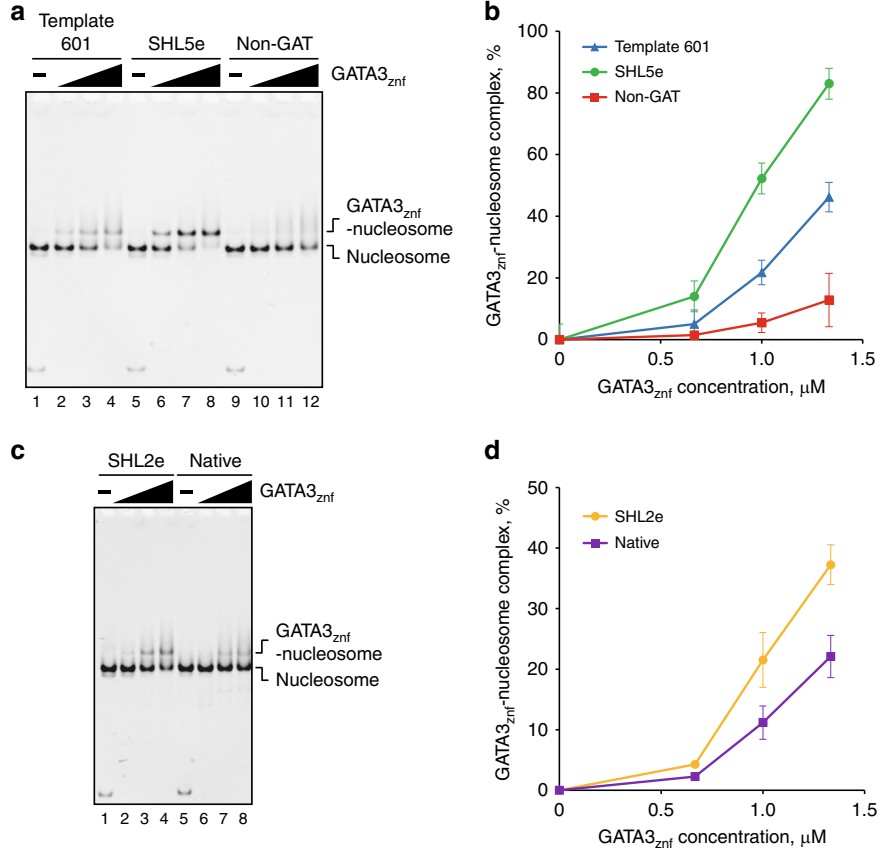

**Fig. 3 GATA3_znf specifically recognizes nucleosomal target DNA. a** EMSA of the GATA3 target or non-target nucleosomes with GATA3_znf. The nucleosome (0.33 μM) was mixed with GATA3_znf (0, 0.67, 1.0, and 1.33 μM). The reaction mixtures were incubated at 25 °C for 30 min and were then analyzed by native-PAGE. The gel was stained with ethidium bromide. Lanes 1–4, 5–8, and 9–12 indicate results for the nucleosomes containing template Widom 601, SHL5e, and non-GAT, respectively. Three independent experiments were performed and the reproducibility was confirmed (Supplementary Fig. 4a, b). **b** Quantification of the results shown in **a**. The average % values of three independent experiments shown in **a** and Supplementary Fig. 4a, b were plotted against the GATA3_znf concentration. Data are represented as mean values ± SD ($n = 3$ independent biological replicates). Source data are provided as a Source Data file. **c** EMSA of a nucleosome containing a native human genome sequence containing the GATA3 target sites. The nucleosome (0.33 μM) was mixed with GATA3_znf (0, 0.67, 1.0, and 1.33 μM). The reaction mixtures were incubated at 25 °C for 30 min and were then analyzed by native-PAGE. The gel was stained with ethidium bromide. Lanes 1–4 and 5–8 indicate results for the nucleosomes containing SHL2e and native sequence with GATA sites around SHL2, respectively. Three independent experiments were performed and the reproducibility was confirmed (Supplementary Fig. 4c, d). **d** Quantification of the results shown in **c**. The average % values of three independent experiments shown in **c** and Supplementary Fig. 4c, d were plotted against the GATA3_znf concentration. Data are represented as mean values ± SD ($n = 3$ independent biological replicates). Source data are provided as a Source Data file.

differences between binding near the dyad axis as opposed to near the edge of the nucleosome.

The model nucleosome utilized for these studies has unique properties that provide both experimental advantages and potential caveats. We wished to explore whether use of a different DNA fragment would provide similar outcomes in biochemical experiments. Accordingly, we selected a human genomic DNA fragment representative of the productively remodeled G1 class. This fragment contains a GATA3-binding motif (5′-AGATTT-CATCT-3′) resembling the sequence used elsewhere in this work positioned near the dyad axis around SHL2 in vivo (Supplementary Fig. 1c upper right panel and Supplementary Fig. 1d). Reconstitution yielded a stable nucleosome with the binding motif localized at position SHL2, which we used for GATA3-binding experiments (Fig. 3c, d and Supplementary Fig. 4c, d). The binding characteristics of the GATA3_znf fragment on this nucleosome were strikingly similar to the binding at SHL2 on the model 601 fragment, suggesting that position on the nucleosome surface, rather than specific DNA sequence of the nucleosome itself, is the dominant factor influencing in vitro DNA binding by GATA3 near the nucleosomal dyad axis.

**Structure of the GATA3_znf–nucleosome complex.** The EMSA revealed that GATA3_znf bound the SHL5d and SHL5e nucleosomes, in which the GATA consensus sequence was inserted around the SHL5.5 position, more efficiently than the other nucleosomes containing the GATA consensus sequence (Fig. 2b, c). Therefore, we prepared the GATA3_znf–nucleosome (SHL5e) complex by the GraFix method in the presence of paraformaldehyde (Supplementary Fig. 2d). The three-dimensional structure of the GATA3_znf–nucleosome complex was reconstructed by cryo-electron microscopy at 3.15 Å resolution (Fig. 4a, Supplementary Fig. 5a–e, and Supplementary Table 1). The GATA3_znf binding did not cause significant structural changes in the histones and the DNA path in the nucleosome (Fig. 4a). In the complex structure, one zinc finger of GATA3_znf was bound to the 5′-GAT-3′ sequence located in the major groove of the SHL5.5 position (Fig. 4a, b upper panel). Intriguingly, a weak, but clear, density was also observed on the 5′-GAT-3′ sequence in the major groove of the SHL6.5 position (Fig. 4b, lower panel). These two 5′-GAT-3′ sequences are separated by six base pairs (Fig. 2a bottom line) and both are accessible to the solvent within the nucleosome (Fig. 4b). The crystal structure of the GATA3 zinc finger–DNA complex is

**a**

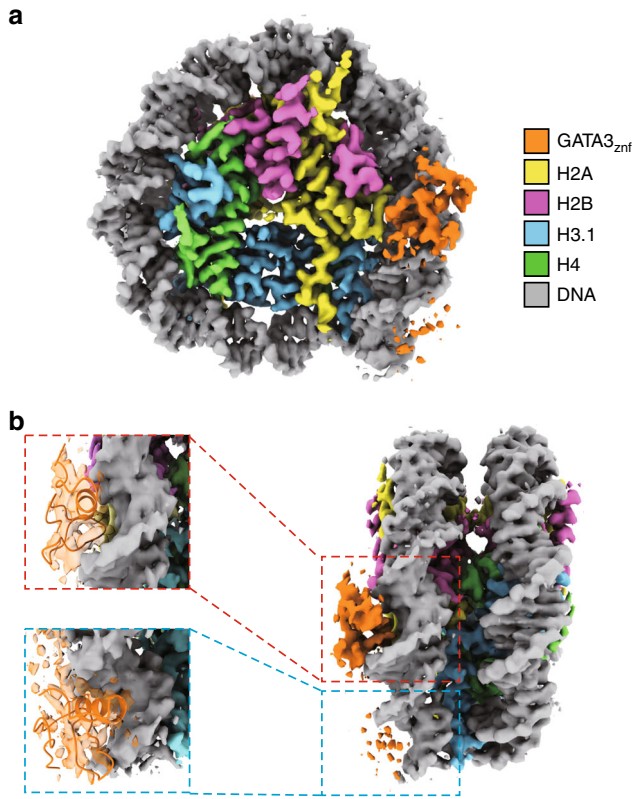

GATA3znf
H2A
H2B
H3.1
H4
DNA

**b**

**Fig. 4 Cryo-EM structure of the GATA3znf–nucleosome complex. a** Cryo-EM structure of the GATA3znf–nucleosome complex at 3.15 Å, contoured at 3.5σ above the mean density. **b** Side view of the cryo-EM structure of the GATA3znf–nucleosome complex (right panel). A close-up view of the SHL5.5 region (dashed red box) is shown in the upper left panel. Crystal structure of the GATA3znf C-terminal zinc finger (PDB ID: 4HCA) was docked into the cryo-EM density map (3.5σ). A close-up view of the SHL6.5 region (cyan dashed line box) is shown in the lower left panel. Crystal structure of the GATA3znf N-terminal zinc finger (PDB ID: 4HCA) was docked into the cryo-EM density map (2.2σ).

fitted very well, when it is superimposed on the cryo-EM map around SHL5.5 of the GATA3znf–nucleosome complex (Supplementary Fig. 6).

**Motifs at SHL5.5 and SHL6.5 are involved in GATA3 binding.** We turned to mutational analysis to assess roles of the three 5′-GAT-3′ elements in the interaction of GATA3 with nucleosomes at SHL5.5 and SHL6.5. We reconstituted nucleosomes with the mutant SHL5e DNAs, in which one of the 5′-GAT-3′ sequences at SHL5, SHL5.5, and SHL6.5 was replaced by a non 5′-GAT-3′ sequence (Fig. 5a). Consistent with the structural data, mutations of the 5′-GAT-3′ sequences at SHL5.5 and SHL6.5 significantly reduced GATA3znf–nucleosome binding, indicating that both these sites function in specific nucleosome binding of GATA3 (Fig. 5b, c and Supplementary Fig. 7). In contrast, the mutation of the 5′-GAT-3′ sequence at SHL5 did not affect GATA3znf–nucleosome binding (Fig. 5b, c and Supplementary Fig. 7). These results confirmed that the two zinc fingers of GATA3znf bind the two 5′-GAT-3′ sequences located in the consecutive major grooves at SHL5.5 and SHL6.5 in the nucleosome.

**Non-random distribution of 5′-GATNxATC-3′ sequences in vivo.** Given the finding that GATA3 binds quantitatively to tandem 5′-GAT-3′ sequences located near the edge of the nucleosome, we returned to our genomic data to assess the

potential for enrichment of such sites at GATA3 ChIP-seq peaks. We considered 5′-GAT-3′ tandem sequences located on the same strand as well as sequences in which 5′-GAT-3′ were located on different DNA strands in a 200 bp window centered on the center of the ChIP-seq peak. We permitted these trinucleotides to be separated by different spacers, ranging from zero to 10 nucleotides (Supplementary Table 2 and Fig. 6). The overall frequency of occurrence of the two types of motifs differ, where loci having both 5′-GAT-3′ motifs on the same strand is approximately twofold more abundant than where the two 5′-GAT-3′ motifs are on opposite strands (Supplementary Table 2). As the probability of GAT motifs on the same strand is two times higher than that on the opposite strand, this result was expected. Compared to randomly selected genomic DNA, we see enrichment for these motifs within the GATA3 ChIP-seq peak centers (Monte Carlo simulation, $p \leq 0.01$), also as expected. In comparison across binding classes, both G1 ($\chi^2$-test, $p \leq 6.9e^{-71}$) and G3 ($\chi^2$-test, $p \leq 5.0e^{-203}$) have higher frequency of occurrence of 5′-GATN$_x$GAT-3′ sites than G2 (Supplementary Tables 2–5). Within the G3 class of sites, this orientation of binding motif is highly enriched with a spacing of six nucleotides between sites ($\chi^2$-test, $p \sim 0$) precisely as observed in the biochemical experiments, while there is less obvious spacing preference in the G1 category (Fig. 6a and Supplementary Table 2). In the 5′-GATN$_x$ATC-3′ orientation, class G1 sites have higher frequency of this orientation than either G2 ($\chi^2$-test, $p \leq 1.7e^{-127}$) or G3 ($\chi^2$-test, $p \leq 2.3e^{-61}$) (Supplementary Tables 2–5). In addition, there is a striking spacing preference in G1 sites for motifs where 5′-GAT-3′ is separated from 5′-ATC-3′ by 3 or 4 bases ($\chi^2$-test, $p \sim 0$). This spacing preference is less obvious, but still significant in G2 ($\chi^2$-test, $p \sim 0$) and G3 sites ($\chi^2$-test, $p \sim 0$ for $N = 4$, $p \leq 4.32e^{-141}$ for $N = 3$) (Fig. 6b and Supplementary Tables 2–5).

**Discussion**
The concept that productivity in nucleation of an enhancer by a pioneer transcription factor differs based on genomic location leads to questions about what distinguishes productive from non-productive loci. In theory, loci within the genome could be distinguished at multiple levels. Higher-order chromatin structure differs as the genome is partitioned by accessibility[20] or by frequency of chromosomal contacts[21] into euchromatin and heterochromatin. It is likely that interaction of pioneers, including GATA3, could differ based on such criteria. At a more local level, the presence (or absence) of chromatin architectural proteins, such as linker histones, could potentially influence outcome of transcription factor/chromatin transactions[22,23]. The details of specific DNA sequence within recognition motifs and how these are displayed on the surface of a nucleosome are expected to influence the physical properties of transcription factor binding (including overall affinity as well as on/off rates) and hence outcome[24–27]. Finally, chromatin and nucleosomes are not static structures in living cells[28–30] and the dynamics of chromatin organization at individual loci likely influence the nature of transcription factor/DNA transactions. These possible influences on outcome at individual loci within the genome are certainly not mutually exclusive and some combination(s) of the above (or other) factors may be critical to the biology.

Here we have investigated specifics of GATA3-binding motif sequence and position relative to local nucleosome structure as possible determinants of outcome of the GATA3/chromatin transaction. We mapped nucleosome boundaries at high resolution within a set of GATA3 bound loci before and after introduction of the transcription factor. These analyses, using the consensus GATA3 motif WGATAR, revealed substantial differences in the relationship between orientation on the nucleosome

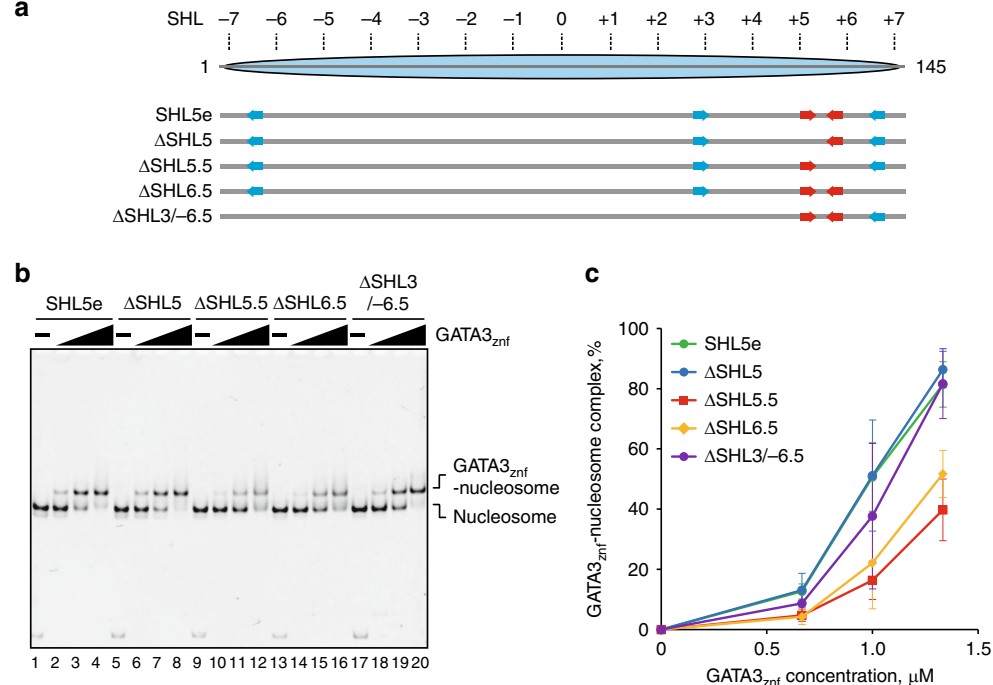

**Fig. 5 GATA3_znf binds the 5′-GAT-3′ sequences at SHL5.5 and SHL6.5. a** Design of the ΔSHL5, ΔSHL5.5, ΔSHL6.5, and ΔSHL3/−6.5 mutants. **b** EMSA of the GATA3 target or mutant nucleosomes with GATA3_znf. The nucleosome (0.33 μM) was mixed with GATA3_znf (0, 0.67, 1.0, and 1.33 μM). The reaction mixtures were incubated at 25 °C for 30 min and were then analyzed by native-PAGE. The gel was stained with ethidium bromide. Lanes 1–4, 5–8, 9–12, 13–16, and 17–20 indicate results with the nucleosomes containing SHL5e, SHL5eΔSHL5, SHL5eΔSHL5.5, SHL5eΔSHL6.5, and SHL5eΔSHL3/−6.5, respectively. Three independent experiments were performed and the reproducibility was confirmed (Supplementary Fig. 7). **c** Quantification of the results shows in **b**. Data are represented as mean values ± SD ($n = 3$ independent biological replicates). Source data are provided as a Source Data file.

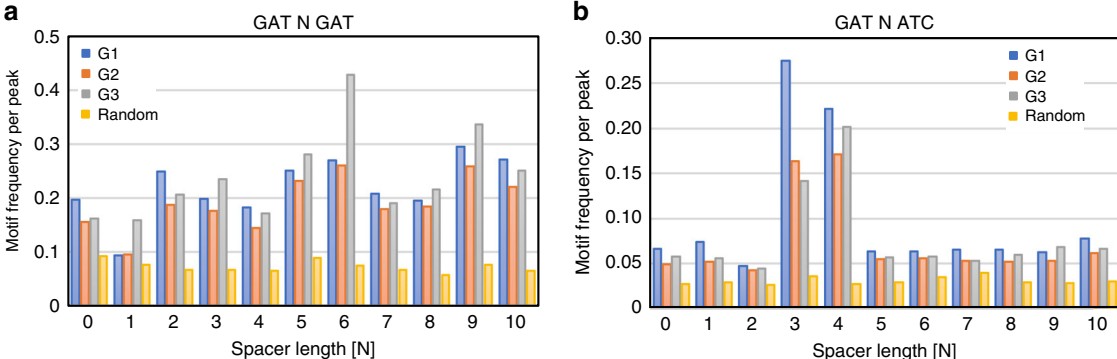

**Fig. 6 Tandem GAT motif enrichment at GATA3 peaks.** The basepair numbers between 5′-GAT-3′ and 5′-GAT-3′ (or 5′-ATC-3′) are indicated on the horizontal axis. **a** GATnGAT frequency in each GATA3 peak group. **b** GATnATC frequency in each peak group. Random genomic regions were selected by using shuffleBed in BEDtools.

surface and outcome. Surprisingly, we observe strong enrichment for GATA3 motifs at position SHL2 in productively remodeled sites. SHL2 is known to be a binding site for multiple chromatin remodelers[31–34], suggesting a potential link between binding site location and transcriptional outcome that is intertwined with the biochemical properties of chromatin remodelers. We also observed preferential positioning of GATA3 near the periphery of the nucleosome after its expression at non-productive loci. A wealth of literature supports the notion that DNA near the periphery of the nucleosome is more accessible for transcription factor binding than more central positions[35–37].

To understand in depth the binding of the zinc fingers of GATA3 to nucleosomal DNA targets, we performed biochemical experiments and determined the cryo-EM structure of the GATA3_znf–nucleosome complex. The biochemical experiments

confirmed that the GATA3_znf can recognize GATA motifs at positions SHL2 and SHL5, as we observed in the genomics data. The cryo-EM structure of the transcription factor/nucleosome complex with GATA3 positioned near the nucleosome periphery indicated that, in the nucleosome, the GATA3 zinc fingers bind the target 5′-GAT-3′ sequences located in consecutive major grooves separated by a minor groove at SHL5.5 and 6.5. This specific geometry avoids steric hindrance between GATA3_znf and nucleosomal histones (Fig. 4). The structure also revealed that the GATA3 binding did not induce a structural change of the histone core and the DNA path in the nucleosome (Fig. 4), although this may be influenced by the spacing and geometry of individual 5′-GAT-3′ sequences found near the nucleosome periphery, where both motifs are solvent accessible. These biochemical and structural data suggest that stable binding near the nucleosome periphery as observed

in loci that fail to nucleate new enhancers may be fundamentally different than interactions occurring near the dyad axis of the nucleosome that are associated with enhancer formation.

In our structure and in the existing crystal structure of GATA3 bound to DNA, both zinc fingers are actively engaged with DNA sequence. In both structures, the transcription factor engages relatively short recognition motifs in agreement with recent studies[24,38,39]. The mutation of one of the two 5′-GAT-3′ sequences bound to GATA3$_{znf}$ substantially reduced the GATA3$_{znf}$ binding affinity, indicating that both 5′-GAT-3′ sequences are required for high affinity GATA3$_{znf}$ binding to the nucleosome (Fig. 5). A 5′-GAT-3′ sequence (SHL5) located in the neighboring major GATA3$_{znf}$ binding site (SHL5.5), but facing the histone surface, did not function as the GATA3$_{znf}$ binding site (Fig. 5). These results demonstrated that the rotational position of the nucleosomal 5′-GAT-3′ sequences is an essential factor for stable GATA binding to the nucleosome periphery.

In contrast to the peripheral position modeled in the cryo-EM structure, our genomic data suggest different spacing and orientation of 5′-GAT-3′ motifs are associated with productive remodeling. In productively remodeled loci, we observe a preference for palindromic GAT sequences with a strong enrichment for spacing of three or four bases (Fig. 6 and Supplementary Tables 2–5). This orientation of 5′-GAT-3′ motifs is seemingly incompatible with productive interaction of GATA3 zinc fingers with both sites while preserving normal nucleosome architecture. At least one major groove 5′-GAT-3′ motif would be adjacent to the octamer surface, not solvent accessible. We propose that transient engagement of both zinc fingers with 5′-GAT-3′ sequences with such spacing and orientation generates strain in the nucleosome structure which increases the probability of a remodeling event. Chromatin remodeling and, ultimately, octamer eviction fundamentally alter the nature of the binding substrate for GATA3, generating a binding motif more akin to naked DNA. The altered kinetics of association and dissociation of the transcription factor from its binding site following such a change are likely to influence recruitment of chromatin modification factors, RNA polymerase and, eventually, nucleation of a new enhancer. This model suggests, somewhat surprisingly, that the orientation of 5′-GAT-3′ motifs to each other and their spacing may play a dominant role in enhancer nucleation by the pioneer transcription factor GATA3 and likely by other members of the GATA family of transcription factors.

This model proposes that steric clash of a dual zinc finger DNA-binding domain with the histone octamer at a key position on the nucleosome surface may underlie the mechanistic basis for chromatin disruption and enhancer nucleation. A very recent publication indicates that a distinct DNA binding domain has both similarities and differences to this model. Cramer and colleagues[40] assessed the interaction of Sox2 and Sox11 DNA-binding domains with a model nucleosome selected biochemically for high affinity interaction of Sox11. Precisely as observed in our genomic analyses, binding near the position SHL2 was preferred[27,40], suggesting that location on the nucleosome surface may be critical to productive binding and chromatin disruption. However, in contrast to GATA3, binding of Sox11 on naked DNA and near the nucleosomal dyad leads to local DNA distortion with widening of the minor groove. This DNA distortion, as opposed to steric clash, leads to displacement of DNA away from the octamer surface[40]. It seems reasonable to conclude that the biophysics of protein/DNA interaction, which differs for the various known DNA binding domains, provides key information on mechanisms by which transcription factors interrupt the chromatin fiber, or fail to do so.

Likewise, our proposal that transcription factor binding events may differ in productivity based on location on the nucleosome surface parallels a recent study by Thoma and colleagues[41]. In this work, nucleosome binding by Oct4 and Sox2 was explored by biochemical selection followed by sequencing. The two transcription factors bound together near the nucleosome periphery at position SHL6 led to DNA distortion under Sox2, with lifting of the DNA from the octamer surface and release of the DNA end from engagement with histones at the entry/exit site. Remarkably, binding the same motif at the mirror image location, SHL + 6, where the relative position of Oct4 and Sox2 to the dyad axis is inverted, led to local distortion but failed to displace the DNA termini from histone engagement. These findings support models depicting location on the octamer surface and relative position of short sequence motifs to each other as integral to chromatin disruption and enhancer nucleation on pioneer transcription factor binding.

In summary, our findings suggest that productive interaction of pioneer transcription factors with chromatin and nucleation of an enhancer can be influenced by both local DNA sequence and local chromatin architecture. It seems likely that preferred binding motif sequence interacts with local nucleosome placement and with the physical nature of protein/DNA interaction to dictate mechanistic details of how nucleosomes are disrupted. Delineation of the details of these events will provide rich information on how pioneer transcription factors invade chromatin and mediate critical biological transitions.

## Methods

**Purification of human histones and the histone octamer**. All human histones (H2A, H2B, H3.1, and H4) were expressed and purified from bacteria[42,43]. For the reconstitution of the histone octamer, four histones (H2A, H2B, H3.1, and H4) were mixed in denaturing buffer (20 mM Tris-HCl pH 7.5, 7 M guanidine hydrochloride, and 20 mM 2-mercaptoethanol), and the mixture was rotated at 4 °C for 1.5 h, followed by dialysis against refolding buffer (10 mM Tris-HCl pH 7.5, 2 M NaCl, 1 mM EDTA, and 5 mM 2-mercaptoethanol). The resulting histone octamer was further purified by Superdex200 (GE Healthcare) gel filtration chromatography.

**Preparation of DNAs**. DNA fragments for nucleosome reconstitution were amplified by PCR and purified by native polyacrylamide gel electrophoresis, using a Prep Cell model 491 apparatus (Bio-Rad). The SHL5e DNA fragment was amplified in *Escherichia coli* cells and was purified for the cryo-EM analysis. The DNA sequences used in this study are derived from the Widom 601 sequence[17] and native human genome sequence. The native human genomic region (GRCh37/hg19 chr4:123,459,211–123,459,355) was selected based on the Capture MNase-seq data (Supplementary Fig. 1d). The DNA sequence of the nucleosomes used in this work are presented in full in Supplementary Table 6.

**Preparation of the nucleosomes**. The nucleosomes were reconstituted by the salt dialysis method[42,43], with the histone octamer and a DNA fragment. The reconstituted nucleosomes were further purified by native polyacrylamide gel electrophoresis, using a Prep Cell model 491 apparatus (Bio-Rad).

**Purification of GATA3$_{znf}$**. The DNA fragment encoding human GATA3 261-371 (Isoform 2; Uniprot ID: P23771-2) was inserted into the pET15b and pGEX-6P-1 vectors. The GATA3$_{znf}$ encoded in the pET15b vector was expressed and purified[44]. The GATA3$_{znf}$ encoded in the pGEX-6P-1 vector was expressed in the *E. coli* BL21 (DE3) codon plus RIL strain (Stratagene), with induction by Iisopropyl β-D-1-thiogalactopyranoside. After the cells were disrupted, the glutathione *S*-transferase (GST)-tagged GATA3$_{znf}$ was purified using Glutathione Sepharose 4B beads (GE Healthcare). The GST-tag was removed by a treatment with PreScission protease. After the GST-tag was removed, GATA3$_{znf}$ was purified by MonoS (GE Healthcare) column chromatography. GATA3$_{znf}$ was further purified by Superdex200 (GE Healthcare) column chromatography with 20 mM Tris-HCl (pH 7.5) buffer, containing 300 mM NaCl, 10% glycerol, 2 mM 2-mercaptoethanol, and 1 μM ZnSO$_4$. The purified GATA3$_{znf}$ was stored at −80 °C.

**Electrophoretic mobility shift assay**. The purified nucleosome was incubated with GATA3$_{znf}$ in 6 μl of reaction buffer, containing 13.3 mM Tris-HCl (pH 7.5), 100 mM NaCl, 6.7% glycerol, 1.3 mM dithiothreitol (DTT), 0.67 mM 2-mercaptoethanol, and 0.3 μM ZnSO$_4$, at 25 °C for 30 min. After the incubation, the samples were analyzed by native-PAGE. The gel was stained with ethidium bromide and imaged with an LAS4000 image analyzer (GE Healthcare) or an Amersham Imager 680 QC (GE Healthcare).

**Sample preparation for cryo-EM analysis**. The SHL5e nucleosome (2.6 μM) was mixed with GATA3$_{znf}$ (6.5 μM), in 240 μl of reaction buffer containing 13.3 mM Tris-HCl (pH 7.5), 100 mM NaCl, 3.3% glycerol, 1.3 mM DTT, 0.67 mM 2-mer-captoethanol, and 0.3 μM ZnSO$_4$. The reaction mixture was incubated for 2 h at 25 °C. After the incubation, the sample was purified and stabilized by the GraFix method[45]. A gradient solution was formed with buffer A (10 mM HEPES-NaOH pH 7.5, 20 mM NaCl, 1 mM DTT, 1 μM ZnSO$_4$, and 5% sucrose) and buffer B (10 mM HEPES-NaOH pH 7.5, 20 mM NaCl, 1 mM DTT, 1 μM ZnSO$_4$, 20 % sucrose, and 3% paraformaldehyde), using a Gradient Master (BioComp). The GATA3$_{znf}$-nucleosome sample was applied onto the top of the gradient solution, and was centrifuged at 89,800 × $g$ (27,000 r.p.m.) at 4 °C for 16 h, using an SW41 Ti rotor (Beckman Coulter). After ultracentrifugation, fractions (0.6 ml each) were collected from the top of the gradient, and were analyzed by native-PAGE. The collected samples were desalted on a PD-10 column (GE Healthcare), equilibrated with 10 mM HEPES-NaOH (pH 7.5) buffer containing 1 mM DTT and 1 μM ZnSO$_4$, and were concentrated with an Amicon Ultra centrifugal filter unit (Millipore).

**Cryo-EM grid preparation and data collection**. For the cryo-EM specimen of the GATA3$_{znf}$-nucleosome complex, 2 μl portions of the sample (0.8 μg DNA/μl) were applied to glow-discharged Quantifoil R1.2/1.3 200-mesh grids. The grids were blotted for 8 s under 100% humidity at 4 °C in a Vitrobot Mark IV chamber (Thermo Fisher Scientific, USA), and were immediately plunged into liquid ethane.
　Images of the GATA3$_{znf}$-nucleosome complex in the vitrified ice were collected using the EPU (Thermo Fisher Scientific, USA) auto acquisition software on a Krios G3i cryo-electron microscope (Thermo Fisher Scientific, USA), operated at 300 kV at a pixel size of 1.05 Å. Images were recorded with 64 s exposure times on a Falcon 3EC (Thermo Fisher Scientific, USA) direct electron detector in the electron counting mode, retaining a total of 51 frames with a total dose of ~50 electron/Å$^2$. The two data collections produced 3,139 images and 2,471 images of the GATA3$_{znf}$-nucleosome complex.

**Image processing**. All movie frames of the GATA3$_{znf}$-nucleosome complex were aligned using MOTIONCOR2[46] with dose weighting. The contrast transfer function (CTF) was estimated by CTFFIND4[47] from digital micrographs with dose weighting. All subsequent image processing of the GATA3$_{znf}$-nucleosome complex was performed with RELION3.0[48]. The particles of the GATA3$_{znf}$-nucleosome complex were picked semi-automatically with a box-size of 180 × 180 pixels, and were processed by two-dimensional (2D) classification followed by 3D classification. The crystal structure of the nucleosome (PDB ID: 3LZ0) was low-pass-filtered to 60 Å, and was used as the initial model for the first round of classification for the first dataset. The 3D map, which contained the GATA3$_{znf}$ density on the nucleosome, was low-pass-filtered to 60 Å, and used as the initial model for the 3D classification for the second dataset. The 3D classes of the GATA3$_{znf}$-nucleosome complex from the two datasets were combined, and 2D classification using 1,026,599 particles was performed to remove junk particles. A second round of 3D classification using the combined dataset containing 760,411 particles was performed, followed by particle polishing and two rounds of CTF refinement. The resolution of the refined 3D map of the GATA3$_{znf}$-nucleosome was 3.15 Å, which was estimated by the "gold standard" Fourier Shell Correlation (FSC) at an FSC = 0.143[49]. The map of the GATA3$_{znf}$-nucleosome was normalized with MAPMAN[50] and visualized with UCSF Chimera[51] and UCSF ChimeraX[52]. Detailed processing statistics values for the GATA3$_{znf}$-nucleosome complex are listed in Supplementary Table 1.

**Capture MNase-seq**. MNase-seq libraries were prepared[11] by washing MDA-MB-231 cells with PBS(−) and treating with a hypotonic buffer (10 mM Tris-HCl pH 7.5, 10 mM NaCl, 3 mM MgCl$_2$, 0.5% Triton X-100) for 15 min on ice. After centrifugation, 4 U of MNase was added to nuclei and incubated for 4 min at 37 °C. Digested DNA fragments were purified using a spin column (Zymo). Mono-nucleosomal fragments were extracted by a QIAGEN gel extraction kit. As a control, the same amount of chromatin was sonicated by COVARIS and the similar length of DNA fragments were purified by the gel extraction kit. Purified (naked) MDA-MB-231 genomic DNAs were also digested by MNase in a same manner and used as another control. Sequencing libraries were prepared by NEXTflex Rapid DNA-Seq kit. Seven hundred and fifty nanograms from each library was used to enrich targeted GATA3 peaks following the instruction in the SureSelect kit (Agilent). Biotinylated RNA probes were hybridized with MNase-seq libraries and enriched by Streptavidin-coupled dynabeads (Thermo). Captured DNAs were further amplified by a 12-cycle PCR.
　The obtained sequence reads were mapped to hg19 genome by bowtie. All paired-end reads were converted to a single fragment, and fragment midpoint and endpoint were extracted for the metaplot analysis. Smoothed nucleosome signals were obtained by iNPS software[16]. Each fragment midpoint was considered as dyad and used to calculate dyad frequency around the GATA3 motif (closest from the highest GATA3 ChIP-seq signals in each peak). The dyad frequency was normalized by the midpoint frequency obtained from capture MNase-seq data in sonicated fragments and plotted on the graph (metaplot). For the heatmap view of motif enrichment analysis, only the fragment midpoints that were located within ±

75 bp from the closest GATA3 motifs were used. The colors were plotted on a log2 scale.

**RNA probes**. RNA probes were obtained from Agilent (Tiling density: 2x, Masking: Least Stringent, Boosting: Balanced). 309 GATA3 peak regions (G1: 109 loci, G2: 100 loci, G3: 100 loci) were selected based on the mappability of the loci. Designed probes cover 1.5 kbp/peak centered on each GATA3 peak center.

**Reporting summary**. Further information on research design is available in the Nature Research Reporting Summary linked to this article.

## Data availability
The data discussed in this publication are accessible through GEO Series accession number GSE72141 and GSE137794. The cryo-EM map of the GATA3$_{znf}$-nucleosome complex is deposited in the Electron Microscopy Data Bank, under accession code EMD-0783. All other relevant data supporting the key findings of this study are available within the article and its Supplementary Information files or from the corresponding authors upon reasonable request. Source data are provided with this paper. A reporting summary for this Article is available as a Supplementary Information file.

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

## Acknowledgements

We are grateful to Y. Iikura and Y. Takeda (University of Tokyo) for their assistance. This work was supported in part by JSPS KAKENHI Grant Numbers JP19K06522 [to Y.T.], JP18H05534 [to H.K.], JP20H00449 [to H.K.], and by the Intramural Research Program of the National Institute of Environmental Health Sciences, NIH (ES101965 to P.A.W.). This work was also partly supported by the Platform Project for Supporting Drug Discovery and Life Science Research (BINDS) from AMED, under Grant Numbers JP20am0101076 [to H.K.] and JP20am0101115j0004 (support number 80) [to M. Kik-kawa to support cryoEM data collection], and the JST ERATO Grant Number JPMJER1901 [to H.K.]. H.T. was supported by a JSPS Research Fellowship for Young Scientists, JP18J13668. M.T. was supported by University of North Dakota Start-up and P20GM104360 from the National Institutes of Health. We gratefully acknowledge expert technical support by the Epigenomics Core, NIEHS.

## Author contributions

H.T. and Y.T. determined the cryo-EM structure of the GATA3$_{znf}$–nucleosome complex. H.T., D.K., and Y.K. performed biochemical analyses of the GATA3$_{znf}$–nucleosome interaction. M.T. performed the genomics experiments. S.A.G. and M.T. analyzed the genomics data. H.K., M.T., and P.A.W. conceived, designed, and supervised the work. All authors contributed to manuscript writing and editing.

## Competing interests

The authors declare no competing interests.
