## [Peer Review File · Nature Communications]

Reviewers' comments:

Reviewer #1 (Remarks to the Author):

In this article the authors follow up on earlier published observations of the GATA3 transcription having three distinct types of binding sites as first revealed by measuring accessibility using ATAC-seq and coupled with GATA3 ChIP-seq. They expand upon this by doing capture MNase-seq to more precisely determine the nucleosome position in these three categories of GATA3 binding sites and validate the earlier observations in that GATA3 either binds to a site already outside of the nucleosome bound region or within a site that is initially bound by a nucleosome that either causes the site to move outside of the nucleosome or to stay within the nucleosome after binding. They find for the nucleosome bound GATA3 that in the two subclasses they have different locations they prefer to bind in the nucleosome. The authors provide biochemical evidence for GATA3 working as pioneer transcription using purified GATA3 and mononucleosomes reconstituted on 601 DNA. They engineered GATA3 binding sites into different positions within the 601 DNA focusing on the SHL5 and SHL2 positions like that observed in vivo and measured the binding affinity of GATA3 for these sites when assembled into nucleosomes. These experiments appear to have been well done, except that in Figure 2 there is not mention of these experiments having been repeated and there is no additional data showing how reproducible these results are. Similar data is also shown in Figure 4 with two replicates mentioned and the other replicate being shown in Supplementary Figure 4 (not Supplementary Figure 3). Unfortunately, the replicate shown does not give the same results as in Figure 4 and instead in Supplementary Figure 4 GATA3 overall appears to have a higher affinity for nucleosomes. There needs to be more replicates performed and the data quantitatively analyzed to confirm the statistical reproducibility of the data in both Figures 2 and 4. It was not possible to effectively compare the differences in GATA3 binding affinity as its site was varied without more repetition and quantitation being done. The fact that they still have GATA3 binding to 601 nucleosomes when all of the GATA3 sites gone is disconcerting and not well addressed in the manuscript. Although it doesn't form a specific band, the general smear nonetheless indicates that GATA3 is binding but not at a specific site. Why does GATA3 bind so well nonspecifically to nucleosomes with an affinity approximately comparable to that with nucleosomes containing GATA3 sites? It wasn't clear in the experiments trying to show that GATA3 prefers to bind to the SHL5 position that the GATA3 binding site at SHL+3 wasn't removed to ensure that it wasn't also contributing.

Using cryo-EM the authors were able to solve the structure of GATA3 bound primarily to the SHL5 and secondarily to the SHL6 positions of nucleosomes to 3.15 angstroms. It was valuable to see this structure and does provide additional insight. Given that GATA3 can also bind at SHL2 it would have been particularly important if they could have solved this structure particularly because GATA3 binding at this position is more likely to perturb the nucleosome than at SHL5 and SHL6. The specific requirements for the spacing between the GAT sites was informative and the different spacing requirement depending on the orientation of the two half sites. I found the statement about the link between chromatin remodelers that bind at SHL2 and GATA3 binding at this same location. Am I correct in think that rather than a link or synergy between them wouldn't this represent more of a competition for the same binding site?

Minor issues

- (1) Line 140 "SHL25b" should probably be SHL5b.
- (2) In Figure 1F the authors did not explain the differences between ellipses with a dashed versus solid line.
- (3) I was wondering why there is a loss of signal at the highest concentration of GATA3 in the gel shift assays?
- (4) Line 667 should read Supplementary Figure 4 and not 3.
- (5) The gel image in Supplementary Figure 2 looked quite skewed or distorted and was wondering why.

Reviewer #2 (Remarks to the Author):

In this study, Tanaka et al purports to study the interaction between GATA3 and nucleosomes. They first defined nucleosomal profiles at GATA3 binding sites before and after GATA3 binding in a system where this factor elicits its pioneer action. In contrast to accessible sites (G2) that are depleted in nucleosomes, both sites that are permissive for pioneer action (G1) and sites that are remodelling-resistant (G3) have nucleosomes at the binding sites before GATA3 binding. The G1 and G3 subgroups present with different organizations of GAT motifs relative to the ChIPseq GATA3 peak center. Using the purified GATA3 zinc finger DNA binding domain, the authors characterized its interaction with reconstituted nucleosomes and found the strongest binding with sites enriched in the remodelling-resistant G3 subset. They used cryo-EM to define the structure of this complex and mutagenesis to validate its interpretation. They finally went back to their GATA3 ChIPseq data to identify GAT motifs enriched around the peaks of the G1-G3 subsets. In agreement with the in vitro nucleosome interaction data, they found enrichment of the direct repeat separated by 6bp in the G3 group but more strikingly observed that palindromic sites separated by 3 or 4 bp are greatly enriched in the G1 pioneer and G2/G3 groups.

Although very interesting, this is an odd paper that finishes where it should have started, namely analysis of GAT sequences present under GATA3 peaks! Indeed, the most striking observation presented in Fig.5 is enrichment of the palindromic sites at the pioneer-permissive sites. Hence, these sites appear the most interesting in terms of understanding the GATA3 pioneer action. However, the authors did not investigate this in any way! The palindromic nature of these sites suggests dimer interaction and prior crystallography data involving some of the same authors showed GATA3 dimer interaction with DNA, albeit with different spacing between the palindrome half-sites. That makes the observation of the 3/4 bp-spaced palindromes under the GATA3 peaks even more provocative as it suggests that such spacing might be critical for pioneer ability.

Nonetheless, the authors have adequately documented the structure of GATA3 bound at the edge of nucleosomes as present in the remodelling-resistant G3 subset. In summary, this reviewer cannot escape but think that this is an incomplete study that misses the most important aspect of GATA3 interaction with pioneered sites within nucleosomes. It is thus preliminary in this reviewer's opinion.

Specific comments

P.7L.193 Is it really Table 2 that should be referenced or rather Supplementary Table S1?

The Discussion is rather superficial and in good part a rehash of the Results section. None of the points raised above in this review are approached. It would seem natural to extensively discuss the finding of the GAT sequences found under the GATA3 peaks in the discussion of a paper that purports to define the interaction of this pioneer with nucleosomes.

Reviewer #3 (Remarks to the Author):

In this study Tanaka and colleagues describe the interaction between the pioneer transcription factor GATA3 and nucleosomes. The author use high-res nucleosome mapping in human cells to study the relation between the position of GATA motifs on the NCP surface and productive enhancer formation. They find that enhancer formation is productive when GATA binding sites map in proximity of the dyad axis. The authors use Widom601 nucleosomes containing a GATA binding

site to identify biochemically tractable GATA3-nucleosome complexes for cryo-EM analysis. A 3.15Å cryo-EM structure is reported, which contains nucleosome and the GATA3 zinc finger motifs, binding on consecutive major grooves. No significant structural change in the histone octamer or the nucleosomal DNA structure is reported. The authors conclude that productive new enhancer formation might derive from specific spacing of GATA3 recognition sequences on the nucleosomal DNA.

This is a very thorough study and the cryo-EM work is state of the art. However, in my opinion, two issues should be addressed before this study can be accepted for publication.

1. The authors should explain why a 145 bp Widom 601 positioning sequence was selected for biochemical and cryo-EM characterisation. Has the use of a natural nucleosomal DNA sequence containing GATA3 binding sites been attempted - if so what were the hurdles encountered / if not, how can the author justify their choice?

2. Related to the first point. Could there be additional DNA engagement modes for GATA3 on nucleosomal DNA (involving DNA distortions), which are not visible because of the use of the strong positioning Widom601 sequence? This point could be addressed by the authors when, on PAGE 10 line 258-260, they claim: "The structure revealed that the GATA 3 binding did not induce a structural change of the histone core and the DNA path in the nucleosome".

I should stress that none of these are serious concerns, although I believe clarifying these points in the text will be appropriate.

MINOR POINTS

3. Have the authors attempted to characterise the complex in the absence of a crosslinking agent and what was the outcome?

4. What is the estimated concentration for the protein sample used for cryo-EM grid preparation (Line 405 on page 14).

5. It would be good to complement Figure 3B with a supplementary Figure comparing the structure of GATA3zfn on nucleosomal DNA (this cryoEM study), with the crystal structure of the complex with naked DNA.

Reviewers' comments:

Reviewer #1 (Remarks to the Author):

In this article the authors follow up on earlier published observations of the GATA3 transcription having three distinct types of binding sites as first revealed by measuring accessibility using ATAC-seq and coupled with GATA3 ChIP-seq. They expand upon this by doing capture MNase-seq to more precisely determine the nucleosome position in these three categories of GATA3 binding sites and validate the earlier observations in that GATA3 either binds to a site already outside of the nucleosome bound region or within a site that is initially bound by a nucleosome that either causes the site to move outside of the nucleosome or to stay within the nucleosome after binding. They find for the nucleosome bound GATA3 that in the two subclasses they have different locations they prefer to bind in the nucleosome. The authors provide biochemical evidence for GATA3 working as pioneer transcription using purified GATA3 and mononucleosomes reconstituted on 601 DNA. They engineered GATA3 binding sites into different positions within the 601 DNA focusing on the SHL5 and SHL2 positions like that observed in vivo and measured the binding affinity of GATA3 for these sites when assembled into nucleosomes.

Comment1)

These experiments appear to have been well done, except that in Figure 2 there is not mention of these experiments having been repeated and there is no additional data showing how reproducible these results are. Similar data is also shown in Figure 4 with two replicates mentioned and the other replicate being shown in Supplementary Figure 4 (not Supplementary Figure 3).

Reply)

We thank the reviewer for this comment with which we completely agree. In the revised manuscript, we repeated the EMSA assays for three times, and the data were presented as new Fig. 2b and Supplementary Fig. 3. The quantitative data were also provided in new Fig. 2c. Previous Fig. 4 was also repeated three times, and the data were presented as new Fig. 5 and Supplementary Fig. 7 with the quantitative data (Fig. 5c).

Comment2)

Unfortunately, the replicate shown does not give the same results as in Figure 4 and instead in Supplementary Figure 4 GATA3 overall appears to have a higher affinity for nucleosomes. There needs to be more replicates performed and the data quantitatively analyzed to confirm the statistical reproducibility of the data in both Figures 2 and 4.

Reply)

In this revision, we first surveyed different binding conditions to reduce the background binding of GATA3 zinc fingers to nucleosomes without GAT sites. We slightly modified the conditions for gel mobility shift assay (different concentration of glycerol) and performed all binding experiments three times. To do so, we freshly prepared all nucleosomes. In the previous manuscript, protein concentration employed in Fig. 4 and Supplementary Fig. 4 were different from the Fig.2 experiments. To avoid this confusion, in the revised manuscript, we repeated the experiments shown in new Fig. 2, new Fig. 3 (previous Fig. 2c), and new Fig. 5 under the same conditions.

Comment3)

It was not possible to effectively compare the differences in GATA3 binding affinity as its site was varied without more repetition and quantitation being done. The fact that they still have GATA3 binding to 601 nucleosomes when all of the GATA3 sites gone is disconcerting and not well addressed in the manuscript. Although it doesn't form a specific band, the general smear nonetheless indicates that GATA3 is binding but not at a specific site. Why does GATA3 bind so well nonspecifically to nucleosomes with an affinity approximately comparable to that with nucleosomes containing GATA3 sites?

Reply)

Thank you very much for this suggestion. To make the point raised by this reviewer clear, we repeated the EMSA assay three times with the nucleosome with or without GAT site under the slightly modified condition. Under the new conditions, we obtained quantitative data for GATA3 affinity to the specific and non-specific nucleosomes, and conclude that the non-specific nucleosome binding of GATA3 is considerably weaker. These new data are presented in the revised Fig. 3a and b.

Comment4)

It wasn't clear in the experiments trying to show that GATA3 prefers to bind to the SHL5 position that the GATA3 binding site at SHL+3 wasn't removed to ensure that it wasn't also contributing.

Reply)

We constructed the SHL5e nucleosome without +3 GAT and -6.5 sites, and confirmed that these sites are not involved in the specific GATA3 binding to the nucleosome. These new data are presented in new Fig. 5 and supplementary Fig. S7.

Comment5)

Using cryo-EM the authors were able to solve the structure of GATA3 bound primarily to the SHL5 and secondarily to the SHL6 positions of nucleosomes to 3.15 angstroms. It was valuable to see this structure and does provide additional insight. Given that GATA3 can also bind at SHL2 it would have been particularly important if they could have solved this structure particularly because GATA3 binding at this position is more likely to perturb the nucleosome than at SHL5 and SHL6.

Reply)

Thank you very much for this constructive comment. In the revised experiments, we have tried to solve the nucleosome structure with GATA3 at SHL2 site. Although GATA3 binds to the nucleosome at SHL2 site, the complex was not stable enough to obtain the structure. Therefore, technique currently we have may not enough to solve it. However, we tested GATA3 binding to the nucleosome containing a native human genome sequence with GAT sites around the SHL2, and found that GATA3 actually binds the native SHL2 site. These new results are presented in new Fig. 3c and 3d.

Comment6)

The specific requirements for the spacing between the GAT sites was informative and the different spacing requirement depending on the orientation of the two half sites. I found the statement about the link between chromatin remodelers that bind at SHL2 and GATA3 binding at this same location. Am I correct in think that rather than a link or

synergy between them wouldn't this represent more of a competition for the same binding site?

Reply)

This is an important issue and we thank the reviewer for raising it. There are several issues that we can envision at play. First, transcription factors (including GATA3) do not bind DNA in vivo in the manner we draw in models. In fact, there is considerable dynamic equilibrium between bound and unbound states. Gordon Hager and colleagues have performed live cell imaging that suggests productive binding events which lead to RNA production are on the order of 10 seconds or so. With transcription factors undergoing such rapid binding and dissociation, there would be ample time for chromatin remodelers to sample the same genomic space as GATA3. In addition, our previous data suggest (PMID:26922637) that DNA binding by GATA3 is not sufficient to induce sensitivity to transposase when we remove the activation domain. Our model suggests that protein-protein interactions that require the GATA3 activation domain increase local concentration of a chromatin remodeler which would then be in the vicinity of the GATA3 binding site when the protein dissociates from DNA. Of course, this model is speculative and will form the basis for additional molecular, genomic, biochemical and structural biology experiments.

Minor issues

(1) Line 140 "SHL25b" should probably be SHL5b.

Reply)

We corrected it accordingly.

(2) In Figure 1F the authors did not explain the differences between ellipses with a dashed versus solid line

Reply)

There is no specific significance to the dashed versus solid ellipses – they were different merely to readily distinguish more than one potential nucleosome position. We made all the nucleosomes with solid lines.

(3) I was wondering why there is a loss of signal at the highest concentration of GATA3 in the gel shift assays?

Reply)

In the presence of excess amount of GATA3, nucleosome formed aggregates with GATA3 and stacked on the well of the gel. To avoid this, in the revised experiments, we re-performed all gel shift experiments under the appropriate conditions.

(4) Line 667 should read Supplementary Figure 4 and not 3.

Reply)

We corrected it accordingly.

(5) The gel image in Supplementary Figure 2 looked quite skewed or distorted and was wondering why.

Reply)

In the presence of paraformaldehyde, the gel image is consistently distorted. This is usual case.

Reviewer #2 (Remarks to the Author):

In this study, Tanaka et al purports to study the interaction between GATA3 and nucleosomes. They first defined nucleosomal profiles at GATA3 binding sites before and after GATA3 binding in a system where this factor elicits its pioneer action. In contrast to accessible sites (G2) that are depleted in nucleosomes, both sites that are permissive for pioneer action (G1) and sites that are remodelling-resistant (G3) have nucleosomes at the binding sites before GATA3 binding. The G1 and G3 subgroups present with different organizations of GAT motifs relative to the ChIPSeq GATA3 peak center. Using the purified GATA3 zinc finger DNA binding domain, the authors characterized its interaction with reconstituted nucleosomes and found the strongest

binding with sites enriched in the remodelling-resistant G3 subset. They used cryo-EM to define the structure of this complex and mutagenesis to validate its interpretation. They finally went back to their GATA3 ChIPseq data to identify GAT motifs enriched around the peaks of the G1-G3 subsets. In agreement with the in vitro nucleosome interaction data, they found enrichment of the direct repeat separated by 6bp in the G3 group but more strikingly observed that palindromic sites separated by 3 or 4 bp are greatly enriched in the G1 pioneer and G2/G3 groups.

Comment1)

Although very interesting, this is an odd paper that finishes where it should have started, namely analysis of GAT sequences present under GATA3 peaks! Indeed, the most striking observation presented in Fig.5 is enrichment of the palindromic sites at the pioneer-permissive sites. Hence, these sites appear the most interesting in terms of understanding the GATA3 pioneer action. However, the authors did not investigate this in any way! The palindromic nature of these sites suggests dimer interaction and prior crystallography data involving some of the same authors showed GATA3 dimer interaction with DNA, albeit with different spacing between the palindrome half-sites. That makes the observation of the 3/4 bp-spaced palindromes under the GATA3 peaks even more provocative as it suggests that such spacing might be critical for pioneer ability.

Reply)

We thank the reviewer this important comment. We initiated our sequence search using the typical position weight matrix identified from ChIP-seq experiments. From ChIP-seq, the PWM for GATA3 is WGATAR – this is what we searched for. The algorithms utilized for defining enriched binding sites within ChIP-seq data typically cannot effectively deal with variable spacing between two sequence elements. We appreciate, in retrospect, that searching for WGATAR may have been simplistic. We agree that the striking occurrence of palindromic sequence near the nucleosomal dyad is likely very important to outcome. We point out that all nucleosome binding experiments in the manuscript using Widom601 based nucleosomes have the GAT sequences on opposite strands with a spacing of 3 bp - as was reported in the crystal structure of GATA3 ZnFingers bound to DNA (PMID23142663) which none of us were

involved in. In fact this paper demonstrated that each GATA3 ZnFinger interacted with a GAT motif – not that the protein bound DNA as a dimer. We are unclear on how to reply to this portion of the comment other than to say that the structural biology literature and our own structure do not support the hypothesis that closely approximated GAT sites are bound by separate molecules of GATA3.

We completely agree, however, that the genomic data warrant further discussion. We have expanded the discussion to address potential models addressing palindromes, spacing and productive remodeling.

Comment2)

Nonetheless, the authors have adequately documented the structure of GATA3 bound at the edge of nucleosomes as present in the remodelling-resistant G3 subset. In summary, this reviewer cannot escape but think that this is an incomplete study that misses the most important aspect of GATA3 interaction with pioneered sites within nucleosomes. It is thus preliminary in this reviewer's opinion.

Reply)

Thank you very much. We have addressed this reviewer's comments, as described below.

Specific comments

1)

P.7L.193 Is it really Table 2 that should be referenced or rather Supplementary Table S1?

Reply)

We thank the reviewer for pointing out that we incorrectly labelled the table in question. We now include this table (which we believe is quite informative) in the main portion of the manuscript as Table 2.

2)

The Discussion is rather superficial and in good part a rehash of the Results section. None of the points raised above in this review are approached. It would seem natural to extensively discuss the finding of the GAT sequences found under the GATA3 peaks in the discussion of a paper that purports to define the interaction of this pioneer with nucleosomes.

Reply)

We agree with portions of the comment. As noted above, we agree that the discussion should address the GAT sequences and productivity. This topic is now the closing paragraph of the manuscript.

We do not entirely agree with the reviewer comments that the remainder of the discussion is rehash of results.

Paragraph 1 defines the potential features of genomic space that we envision may influence productivity of a pioneer.

Paragraph 2 of discussion links enrichment of sites near the nucleosomal dyad with the known biochemical properties of chromatin remodelers.

Paragraph 3 summarizes the biochemistry of nucleosome binding and suggests that binding near the periphery and near the dyad may be different in outcome.

Paragraph 4 and new paragraph 5 discuss the details of the two scenarios – binding near the periphery and near the dyad.

Reviewer #3 (Remarks to the Author):

In this study Tanaka and colleagues describe the interaction between the pioneer transcription factor GATA3 and nucleosomes. The author use high-res nucleosome mapping in human cells to study the relation between the position of GATA motifs on the NCP surface and productive enhancer formation. They find that enhancer formation is productive when GATA binding sites map in proximity of the dyad axis. The authors use Widom601 nucleosomes containing a GATA binding site to identify biochemically tractable GATA3-nucleosome complexes for cryo-EM analysis. A 3.15Å res cryo-EM structure is reported, which contains nucleosome and the GATA3 zinc finger motifs, binding on consecutive major grooves. No significant structural change in the histone octamer or the nucleosomal DNA structure is reported. The authors conclude that

productive new enhancer formation might derive from specific spacing of GATA3 recognition sequences on the nucleosomal DNA.

This is a very thorough study and the cryo-EM work is state of the art. However, in my opinion, two issues should be addressed before this study can be accepted for publication.

1. The authors should explain why a 145 bp Widom 601 positioning sequence was selected for biochemical and cryo-EM characterisation. Has the use of a natural nucleosomal DNA sequence containing GATA3 binding sites been attempted - if so what were the hurdles encountered / if not, how can the author justify their choice?

Reply)

We thank the reviewer for this useful and insightful comment. Indeed, it has been a concern of ours for some time. We believe addressing this comment has improved our manuscript.

We initially employed the Widom 601 sequence because it has been extensively characterized and is well behaved biochemically.

In this revision, we have included one native human nucleosome of the productively remodeled G1 class. The binding site is very similar to the sequence we introduced into the Widom nucleosome. We find no substantial difference in GATA3 binding with this nucleosome. This data and relevant explanation to illuminate the general readership of Nature Communications is now presented on pages 6 and 7 and in Figure 3c, 3d, Supplementary Fig. 1c, 1d, Supplementary Figure 4c, 4d.

2. Related to the first point. Could there be additional DNA engagement modes for GATA3 on nucleosomal DNA (involving DNA distortions), which are not visible because of the use of the strong positioning Widom601 sequence? This point could be addressed by the authors when, on PAGE 10 line 258-260, they claim: "The structure revealed that the GATA 3 binding did not induce a structural change of the histone core and the DNA path in the nucleosome".

I should stress that none of these are serious concerns, although I believe clarifying these points in the text will be appropriate.

Reply)

Thank you very much for this comment as well.

We believe the answer may be complicated. Near the nucleosomal dyad (SHL2 variants) the binding properties seem different than near the periphery (SHL5.5/6.5), likely owing to the utilization of the fortuitous GAT motif in 601 DNA which is spaced so that GAT sites are in consecutive, solvent accessible major grooves. We do not know whether the more compact binding motifs used elsewhere have the same properties – and in fact they may not (see new last paragraph of discussion). We are in full agreement with this reviewer that sorting these details out is an important priority – but we believe it to be beyond the scope of this manuscript. We are currently planning the next generation binding experiments to specifically address this very issue.

We propose changing the text in discussion to read: “The structure also revealed that the GATA3 binding did not induce a structural change of the histone core and the DNA path in the nucleosome (Fig. 4), **although this may be influenced by the spacing and geometry of individual 5'-GAT-3' sequences found near the nucleosome periphery, where both GAT motifs are solvent accessible.**” in the revised text on page 11.

MINOR POINTS

3. Have the authors attempted to characterise the complex in the absence of a crosslinking agent and what was the outcome?

Reply)

GATA3 binding to the nucleosome is not strong enough for cryo-EM analysis without crosslinking agent.

4. What is the estimated concentration for the protein sample used for cryo-EM grid preparation (Line 405 on page 14).

Reply)

We described it in the Methods section accordingly.

5. It would be good to complement Figure 3B with a supplementary Figure comparing the structure of GATA3zfn on nucleosomal DNA (this cryoEM study), with the crystal structure of the complex with naked DNA.

Reply)

Thank you very much. In the revised manuscript, we present comparison between cryo-EM and crystal structures of GATA3-DNA or nucleosomal DNA, in new Supplementary Fig. 6.

REVIEWERS' COMMENTS:

Reviewer #1 (Remarks to the Author):

The authors have adequately addressed my previous questions and concerns. It is unfortunate that they were unsuccessful at obtaining structures of GATA3 bound to its cognate site at the SHL2 positions given their data suggesting this is likely to be the more biologically relevant in terms of enhancer function. It will also be important for the authors to note the recent publication from Patrick Cramer's lab on the structure of two other pioneer factors, SOX2 and SOX11, bound to nucleosomes at the SHL2 position published last month in the journal Nature. Cramer's data supports several aspects mentioned at the end of their discussion. The subject of their article is very timely and will be of broad interest.

Reviewer #2 (Remarks to the Author):

The authors have adequately revised the manuscript in response to this reviewer's comments. The extended discussion finishes with the challenging biological question of sequence requirements for pioneer action.

Reviewer #3 (Remarks to the Author):

In the revised version of their manuscript the authors have addressed all of my comments satisfactorily. I believe this work should be accepted for publication in Nature Communications.

Detailed response to reviews

Reviewer #1 (Remarks to the Author):

The authors have adequately addressed my previous questions and concerns. It is unfortunate that they were unsuccessful at obtaining structures of GATA3 bound to its cognate site at the SHL2 positions given their data suggesting this is likely to be the more biologically relevant in terms of enhancer function. It will also be important for the authors to note the recent publication from Patrick Cramer's lab on the structure of two other pioneer factors, SOX2 and SOX11, bound to nucleosomes at the SHL2 position published last month in the journal Nature. Cramer's data supports several aspects mentioned at the end of their discussion. The subject of their article is very timely and will be of broad interest.

We thank the reviewer for the positive comments and the conclusion that our article is very timely and will be of broad interest.

We also agree wholeheartedly with this reviewer that discussing the similarities/differences between our paper and the Cramer work (as well as very similar work from Nicolas Thoma) would add to the depth of the manuscript. We have added additional discussion to address this request. We thank the reviewer for the suggestion.

Reviewer #2 (Remarks to the Author):

The authors have adequately revised the manuscript in response to this reviewer's comments. The extended discussion finishes with the challenging biological question of sequence requirements for pioneer action.

We thank the reviewer for their support of this version and for the astute comments to the previous version that helped us improve the manuscript and deal with an important issue.

Reviewer #3 (Remarks to the Author):

In the revised version of their manuscript the authors have addressed all of my comments satisfactorily. I believe this work should be accepted for publication in Nature Communications.

We thank the reviewer for their support and positive comments.